# Do We Drop the Ball When We Measure Ball Skills Using Standardized Motor Performance Tests?

**DOI:** 10.3390/children9030367

**Published:** 2022-03-06

**Authors:** Bouwien Smits-Engelsman, Dorothee Jelsma, Dané Coetzee

**Affiliations:** 1Department of Health & Rehabilitation Sciences, Faculty of Health Sciences, University of Cape Town, Cape Town 7925, South Africa; 2Physical Activity, Sport and Recreation, Faculty Health Sciences, North-West University, Potchefstroom 2520, South Africa; dane.coetzee@nwu.ac.za; 3Clinical and Developmental Neuropsychology, University of Groningen, 9712 TS Groningen, The Netherlands; l.d.jelsma@rug.nl

**Keywords:** children, ball skills, construct validity, motor performance test, MABC-2, BOT-2, PERF-FIT

## Abstract

Background: Ball skill performance changes over time during childhood and depends on the child’s physical and psychological characteristics, environmental opportunities, and task constraints. The aim of this study was to examine whether different standardized ball skill-items measure similar constructs and whether results differentiate among age groups. *Methods*: A total of 250 children (6–10 years) were administered the Movement Assessment Battery for children second edition (MABC-2), the subtest upper-limb coordination of the Bruininks–Oseretsky Test of Motor Proficiency second edition (BOT-2), the tennis ball throw of the Fitness Test Battery (FTB), and the ball skills and overhead throw of the Performance and Fitness test (PERF-FIT). *Results*: Correlations among scores of BOT-2, FTB, and PERF-FIT ranged from fair to moderate, but correlations were low with the MABC-2. Principal component analysis retrieved three components: the first component contained catch, dribble, bounce, and throw items (MABC-2, BOT-2-SF, and PERF-FIT, respectively); the second contained throwing for distance (PERF-FIT and FTB); the third contained aiming (MABC-2). Most ball skill scores differed significantly between 6–7 and 7–8 years, but less between 8–9 years; the PERF-FIT discriminated best between age groups. *Conclusion*: Most ball skill items in motor tests belong to a comparable construct containing throw, dribble, bounce, and catch skills. However, aiming tasks belong to a separate category. Additionally, throwing for distance is a task that requires predominantly explosive power.

## 1. Introduction

The mastery of certain fundamental motor skills is a prerequisite for daily life functioning and participation in physical or sport-specific activities [1]. Barnett and collaborators showed that fitness, physical activity, and sport competence perception were positively associated with object control or ball skills [2]. Most school children enjoy playing ball games and learn the fundamentals of these skills effortlessly. Practicing ball skills leads to gradual adjustments, resulting in better performance. This way, children become increasingly skilled with age and experience [3,4,5]. The emerging competency is the result of organismic (the individual’s physical and psychological characteristics), environmental (the external forces surrounding the performer, opportunities for physical education and sports), and task-related constraints (the rules and goals of the task and the equipment used) [6].

How ball skill performance changes over time and which of these constraints determine progression in ball skill performance is one of the questions in developmental motor control studies. Knowledge, about which constraints underlie ball skill performance and its improvement over time, is of crucial importance for the development of intervention programs [7].

One of the problems in understanding changes in ball skill performance is the large set of activities that make up “ball skills” and the many degrees of freedom in the human motor system involved in the control of successful performance. Due to an absence of consensus about which ball skill activities comprise competency in ball skills, various motor tests assess different ball skill activities (aiming, throwing over distance, catching a ball thrown by a tester or by subject via a wall, bouncing a ball, hitting a ball, kicking a ball). These standardized test items are often simplified tasks compared to skills needed in games. The level of ball skill proficiency in games and sports is highly dependent on being on the right place at the right time and then making a targeted throw or a secure catch.

Nonetheless, standardized tests are important to assess the level of competency in ball skills, as well as to assess progress when a child gets older or after intervention in the case of developmental disorders. Clinicians often report a child as having poor ball skills if they score below the fifth percentile on a standardized test. An important question is whether ball skill items in currently available motor tests measure similar constructs or examine different, unrelated constructs, since often only low to moderate correlations are found among ball skill items of different motor tests [1,8]. This might imply heterogeneity in the operationalization of “ball skills” as a construct.

Information about which constructs are measured in ball skill test items is important in pediatric physical therapy and adapted physical education. One of the changes in the intervention for developmental disorders in last decennia has been a shift from an underlying deficit framework for evaluation and training toward task analysis and task-oriented training of functional activities based on a dynamic system- or constraint-based approach [9,10,11]. This view on motor learning is related to the notion of generality versus specificity of motor learning [12,13].

In the case of generality, it would mean that the different ball skills are highly correlated, implying a general underlying factor, which measures the same construct with one large principal component that would explain the majority of the variance (plus some smaller residual factors) [14]. In clinical practice, this would mean that, if a child is trained on basketball and tennis skills, they would also get better at table tennis because this skill uses the same underlying qualities, processes, or elements, such as predicting the ball trajectory, anticipating where to be when, and generating a coordinated arm movement [15].

On the other hand, if learning a ball skill is very task-specific, the communality (correlation) and transfer between tasks is expected to be low. If one would predict the principal components from this perspective, ball skill tasks that are the most similar, according to their task constraints (and not underlying processes), are expected to cluster. For instance, predicting the ball trajectory is very different for a basket, tennis, and ping pong ball, as are their bouncing behaviors. This makes the task variable that predicts where the player should be at a specific time dissimilar between the tasks. Additionally, there are three different ball-handling patterns to be generated to play the ball, using bare hands, bat, or racket.

Knowledge of whether ball skill items cluster together as one unifying construct or are made up of separate clusters of items is important not only from an assessment perspective but also from an intervention perspective. If ball skill items belong to different clusters of tasks, one would expect transfer between task clusters to be low. This would imply that tasks belonging to different clusters should be specifically trained during intervention to improve performance (task specificity).

Therefore, the primary goal of this study was to examine the communality in the different ball skill items to confirm the construct of “ball skills” they intend to measure. The ball skill items of the most often used motor tests worldwide, the Movement Assessment Battery for Children [16] and the Bruininks–Oseretsky Test for Movement Proficiency [17], of a newly developed motor test, the PERF-FIT [18], and of the Physical Fitness Test Battery for Children [19] were used for this purpose.

In addition, ball skill performance is known to improve with age and, thus, with experience, practice, and general development [20,21]. To capture developmental changes in ball skill performance, the items of a motor test assessing ball skills should show age-related changes in performance. Therefore, our secondary goal was to examine whether performance on the ball skill items improves as a function of age and if test items differentiate among age groups.

## 2. Materials and Methods

### 2.1. Participants

In total, the results of 250 children were included in the analyses (6 years: *n* = 46; 7 years *n* = 43; 8 years *n* = 101; 9 and 10 years *n* = 60) (Figure 1). Demographic characteristics of this randomly selected group of children are summarized per MABC-2 age band in Table 1.

### 2.2. Assessment Tools

#### 2.2.1. Movement Assessment Battery for Children Second Edition (MABC-2)

All children completed MABC-2 [16] age band 1 (6 year old children) or 2 (7 to 10 year old children). The test contains eight items for each age band, divided into three components: manual dexterity, ball skills, and balance. The raw item scores were converted to standard scores to classify motor performance of the participants. The MABC-2 test is considered to have good psychometric properties [16,22,23,24]. The aiming and catching items were used as items in the ball skill analysis. The items of the MABC-2 change with age; however, given that the items of the two age bands have the same scale (10 throws and 10 catches) and are supposed to measure the same construct, raw scores were used in the comparison to the ball items of the other tests.

#### 2.2.2. Bruininks–Oseretsky Test of Motor Proficiency 2nd Edition (BOT-2)

The BOT-2 [17] is a normative test that assesses fine and gross movement skill development in children from 4 to 21 years of age. BOT-2 has excellent inter-rater reliability (*r* = 0.97), good test–retest reliability (*r* = 0.85), and good construct validity [25]. For this study, we used point scores of the items 1 and 6 of the upper-limb coordination of the short form of the BOT-2 (*n* = 250) and in a subsample (*n* = 88) of values for all 7 items for the upper-limb coordination subtest.

#### 2.2.3. The Fitness Test Battery for Children (FTB)

The FTB developed by Fjørtoft and coauthors [19] is a test that focuses on common activities included in most children’s everyday play activities. This fitness test includes one functional standardized ball skill item, which was used in this study. In this item, the child throws a tennis ball with one hand (the child chooses which hand) as far as possible. The child stands with the contralateral foot just behind a starting line. The child is not supposed to step over the line. Two test trials are performed with 15 s rest in between trials. Distance in meters is measured between the starting line and the landing point of the ball. The tennis throw item has good reliability (ICC 0.92, CI 0.83–0.97) [19].

#### 2.2.4. Performance and Fitness Test (PERF-FIT)

The PERF-FIT is a recently developed standardized test battery to assess motor skill related fitness in children 5–12 years of age [18]. The PERF-FIT is the first standardized test to have norms for African children. Access to affordable testing tools is a serious consideration when measuring motor skills in low-resourced areas. The test has no specific space requirements (outside or inside) and is suitable for this age group (elementary school children). The items of the PERF-FIT were designed to be used over the full age range. The PERF-FIT has good structural and ecological validity, excellent content validity, and good reliability [26,27,28].

The test has two subscales: an agility and power subscale and a motor skill performance subscale. The motor skill performance subscale contains five skill item series of increasing difficulty: bouncing and catching, throwing and catching, jumping, hopping (left and right), and balance. All children start at the easiest skill level, and a series is terminated when they do not reach the criterion number of points for the item after two trials. For this study, only the bouncing and catching, and throwing and catching items were selected.

The power and agility subscale contains five items: running, stepping, side jump, long jump, and overhead throw. For the agility and power subscale, children perform two trials for each item, and the best score is recorded. From this subscale, only the overhead throw with a 2 kg sandbag was selected.

### 2.3. Procedure

First, we obtained permission from the school district to approach the head teachers. Next, the study purpose, test procedures, benefits, and risks were explained to parents. All children between 6 and 10 years of age, in the classes invited for this study, were eligible to participate after parents or caretakers signed consent and after the children gave assent. Children whose caregivers answered in the affirmative on any of the questions of the children’s Physical Activity Readiness Questionnaire (PAR-Q) were excluded from the study [29]. In addition to MABC-2, BOT-2, FTB, and PERF-FIT scores, data collected included age, height, weight, and gender.

To avoid exceeding the maximum time allowed by the schools, in one school, children were tested on all seven BOT-2 upper-limb coordination items (*n* = 88), while the children from the other schools performed the FTB tennis ball throw for the distance item (*n* = 172) (see Figure 1).

The lead author trained all the assessors (senior researchers and postgraduate students with a qualification in human movement science, specializing in kinderkinetics, physiotherapists specializing in pediatrics) for at least 8 h on all measurement tools.

Assessments took place at the school’s premises outside, divided over at least two sessions. Total test time varied between 90 and 120 min depending on the skill level of the children. Children with lower motor proficiency more often needed two trials. All tests were administered according to the test manual guidelines. Most of the children wore school uniforms during testing.

### 2.4. Data Analysis

Data were checked for normality using the Kolmogorov–Smirnov test, and distribution of the data of the ball skill variables was not Gaussian, except for PERF-FIT overhead throw.

Descriptive statistics including mean, standard deviation, median, minimum, and maximum values were used to summarize the data. Spearman’s rank-order correlation coefficients were computed to examine the relationships among ball skill items of the MABC-2, BOT-2, Fitness Test Battery, and PERF-FIT items. Relationships among the different ball scales were interpreted as follows: little to no relationship (*r* = 0.00–0.24), fair relationship (*r* = 0.25–0.49), moderate to good relationship (*r* = 0.50–0.75), or good to excellent relationship (*r* > 0.75) [30].

As the MABC-2 aiming and catching standard scores were corrected for age and all the other ball scales were not, relationships were calculated using the raw scores of the two MABC-2 ball items (aiming and catching). Because MABC-2 items change with age, relationships were also calculated separately for the subsample of 6 year old children (*n* = 46) and 7–10 year old children (*n* = 204).

To determine whether the ball skill items measure the same construct, principal component analysis (PCA) with varimax rotation was used. Orthogonal factor scores were derived on the basis of a correlation matrix, with a minimum eigenvalue for extraction set at 1 to determine the number of dimensions in the ball scales. Screen plots, total variance explained, rotated component matrix, and transformation matrix were investigated. Loadings ≥0.4 per item were considered relevant. All ball scale scores containing raw values (number of caught balls, target hits, and distance thrown) were included for the PCA.

To test for age differences, the effect of age group on performance of the ball skill items was assessed using a Kruskal–Wallis test. Nonparametric post hoc tests were performed to compare differences between two adjacent age groups. Age groups 9 and 10 were combined because of the lower number of 10 year old children.

Data analyses were performed using SPSS 28 [31]. The level of significance was established at *p* ≤ 0.05.

## 3. Results

### 3.1. Relationships among Ball Scales

First, to calculate the correlation among ball scales, all data were used (see Table 2). Fair to good relationships were found between PERF-FIT and BOT-2 items (*r* = 0.37–0.62), as well as among PERF-FIT, BOT-2, and overhand tennis throw (*r* = 0.23–0.50). Overhand tennis throw, PERF-FIT, and BOT-2 skill items showed little to no relationship (*r* = 0.03–0.22) with MABC2 raw ball items scores and little to fair values with the MABC2 catch standard scores (r = 0.09–0.33). Relationships between MABC2 aiming standard scores and the other test items were not significant (see Table 2).

In the next step, correlations among BOT-2, FTB, and PERF-FIT items were calculated separately for age bands 1 and 2 of the MABC-2 (see Table 3); the results showed that the pattern of correlation was different for each age band.

Lastly, Table 4 shows the relation among ball scales in the sub sample of children who performed all upper-limb items of the BOT-2. The highest relationship was found between the dribbling item of the BOT-2 and the bouncing and throwing items of the PERF-FIT (*r* = 0.68 and 0.69).

### 3.2. Dimensionality of Ball Skills

Because the MABC-2 has different items for age bands 1 and 2, in the PCA, only data of the children in age band 2 (7–10 years old) were used. In the analysis including two ball skill items of MABC-2, two items of upper-limb coordination scores of BOT-2-SF, FTB tennis throw, and total scores of bounce and catch, and throw and catch items of the PERF-FIT, three factors with an eigenvalue of 1 emerged explaining 65.4% of the variance (Kaiser–Meyer–Olkin 0.77). The first component contained the MABC-2 catch item, the BOT-2-SF catch and dribble items, and the PERF-FIT bounce and catch, and throw and catch total scores. The second component showed a cluster of the throwing items requiring explosive power: overhead throwing of the sandbag and tennis ball. The MABC-2 aiming item loaded on the third factor, showing that aiming is a different skill (see Table 5).

### 3.3. Ball Scale Performance and Age

The performance of children on all ball items of the BOT-2 and PERF-FIT was significantly different between age groups (Table 6 and Figure 2). Post hoc tests showed that the performance on the three PERF-FIT items was different across each increasing year except for bounce between 7–8. For the BOT-2 item 1 (drop and catch two hands), 6 year old children performed significantly more poorly than the older children, whereas, for BOT-2 item 6 (alternating dribble), scores were different across all age groups. For the FTB item, the distance thrown for 7 and 9 year old children was significantly different, but not between adjacent age groups (Figure 3). MABC-2 raw score age band 2 items were different across age groups. The post hoc test for the raw score showed differences between 7 and 8 year old age groups, but scores between 8 and 9/10 year old children were not different (see Table 6 and Figure 2).

## 4. Discussion

The primary goal of this study was to examine the communality in the different ball skill items to explore the construct of “ball skills”. Data showed that test items used in standardized tests that measure different ways to handle a ball (specifically, catch or bounce) largely measure a comparable construct. This confirms the concurrent validity of the BOT-2 and PERF-FIT ball scales. Items that use throwing for distance, as measured by the overhead throw of a 2 kg sandbag or a tennis ball, seem to measure another component, namely, explosive power. Moreover, aiming at a target in the MABC-2 and BOT-2 seems to belong to a separate category, exhibiting a lower relationship with bouncing and catching items.

Concerning our second aim, to test if performance on all ball scales used in standardized tests improved as a function of age group, this was largely confirmed. The sample included children aged 6–10 years, an age range in which ball skills develop [32,33]. As expected, ball skill raw scores increased with age. The PERF-FIT discriminated best between adjacent age groups.

### 4.1. Constructs Measured in the Ball Scales

Both MABC-2 and BOT-2 have one aiming item, and, in both cases, this item was shown to have a low correlation with the bouncing, throwing, and catching items, indicating that the latter represents a different set of skills. It seems that aiming at a target on the floor is only minimally related to throwing, catching, and dribbling. Aiming at a target on the wall also showed a low relationship with the other ball scales. In the case of MABC-2, aiming constituted 50% of the ball performance total score and explained 12.7% of the variance in ball skill outcomes. In the case of BOT, the aiming item was only present in the full-scale BOT-2 and was one of seven items measuring upper-limb coordination.

Both PERF-FIT and BOT-2 use bouncing on the floor. The MABC-2 allows bouncing via the floor for 7 and 8 year old children, who try to catch a ball that bounces back from the wall on the floor with two hands, while 9 and 10 year old children need to catch with two hands without bounce. This change in the catch item leads to a large increase in difficulty, which is corrected for by the standard scores available per ball item per age year. Interestingly, the PERF-FIT is the only test that contains bouncing, throwing, and catching with the non-preferred hand in this age range. In the MABC-2, one-handed catching is only examined in older children (11–16 years old), whereas, in the BOT-2 short form, it is present as alternations between the preferred and non-preferred hand. The PERF-FIT items that involve clapping hands are clearly more difficult but did not constitute a separate factor, as might have been expected, because they consist of combining a sequence of different tasks (bouncing, clapping, and catching or throwing, clapping, and catching).

Throwing for distance seems, for a large part, to be determined by explosive power and has less in common with aiming skills. Aiming and distance throwing require a different movement pattern depending on the goal. Aiming is mainly characterized by accuracy constraints. Throwing for distance also requires a different movement pattern (such as stepping with the contra lateral leg and the follow through of the arm movement), which becomes an important factor for throwing a larger distance.

### 4.2. Ball Skill Competency

Ball skills are known to be difficult for children with developmental disorders. However, the importance of the level of ball skill competency as a prerequisite for active participation in play is often overlooked [34]. Ball skills require, in addition to control of the ball, planning of concurrent and sequential actions, which seem to improve when ball skills improve [35]. For example, to participate in a game of basketball, a child needs to combine running, catching, dribbling, throwing, and jumping while throwing. Solving the unpredictability by reading your opponent or teammate through recognizing the moves and circumstances is one of the key components of skilled catching and throwing at a level of being successful to participate in games or sports. Thus, many factors affect catching success, which cannot be evaluated by looking at the outcomes of standardized norm-referenced tests. Numerous task constraints, such as ball size, bouncing properties, speed, trajectory, distance, and height of interception point, can make the task simple or complex [36,37,38,39,40].

Importantly, to better standardize test items, developers choose many constraints; the child stands behind a line, is warned before the tester throws a well-aimed ball, or receives a ball from a specific distance and at one height. By aiming for standardization of ball skill items, we moved away from real-world ball skill requirements where children need to be sensitive to the perceptual aspects of the throw or catch and respond with an appropriate body orientation, hand choice, and planned movement to intercept on the basis of visual information about the flight of the ball [41,42,43,44]. More complex actions that occur rapidly lack regularity and predictability differentiate best between an expert level and lower level of ball skills [45]. However, motor tests assess ball skills in a very predictable context, which make the tests reliable but more distant from real-world ball games where trajectory prediction is one of the determining factors [46]. Items in which children project the ball they must catch themselves seem to be closer to real-world skills because they include at least an agility component as opposed to more static test situations (standing on a mat or behind a line). As the task becomes more complex, prediction and adaptation become more difficult, and processes of executive functioning are needed, such as attention, working memory, planning, and problem solving [34,47].

In summary, the throwing, bouncing, and catching items in the standardized tests seem to capture the basic development of gross visual–motor coordination associated with ball skills in children aged 6–10 years. However, many aspects of throwing and catching are not tested in standardized motor tests. For instance, throwing the ball to a running teammate requires the rightly predicted time and place. The child is constantly required to make appropriate decisions as a function of the changing position of the ball and of other players. Catching a ball thrown to you in the field, therefore, requires the combination of predicting the trajectory of the ball and planning the best way to get to the predicted location in time (without running into an obstacle or opponent) [41,43]. Importantly, the skill of catching provides the opportunity to examine the closely intertwined perceptual and motor aspects in a task that is externally constrained at the spatial and temporal level [45]. Thus, what current ball skill items of standardized test measure is only an approximation of ball skill performance needed in sports and games. Hence, we partly “drop the ball” by aiming for reliable clinical standardized tests. Our analysis showed that test items used in standardized tests largely measure three comparable constructs: (i) catching, throwing, and bouncing, (ii) throwing for distance, and (iii) aiming.

### 4.3. Clinical Implication

Good ball skills are important for physical or sport-specific activities. Although ball skill items measure increased performance over age groups, these test results give too little information to develop an intervention plan. The way we test ball skills in standardized tests is distant from sport-specific activities, in which perception action coupling and anticipatory control will determine success. Thus, the current ball skill items of standardized tests measure only an approximation of ball skill performance needed in sports and recreational games. Additional criterion-based evaluation and extensive task-analysis are needed through changing the constraints, using different balls, thrown at different speeds, from various distances and height. Observing the changes in the nature of the emerging movement patterns will be helpful when designing ball skill training programs for children with movement problems or slower motor development. Once the basic catching and throwing skill is mastered, ball skill training needs to become more dynamic because performance in sport and leisure also involves agility, unpredictable contexts, and perceptual reading of the other players’ behavior. As skill learning progresses, this will result in greater flexibility in the way children move their limbs to successfully accomplish a task. Such adaptability allows skilled performers to execute a task in different initial conditions and changing environmental constraints [48].

### 4.4. Limitations

Participants in this study were from a low-resourced environment, had limited access to organized physical activities, and showed a relatively high frequency of “at risk” motor proficiency on the MABC-2 classification. Thus, this random cross-sectional sample included children over the full range of the motor proficiency spectrum, which adds to the validity of the findings. Moreover, children in the present study did not have structured physical education in school but did have breaks with outside playtime.

Test items in the included standardized tests contain a limited spectrum of ball skills. Hitting and kicking skills were most likely not incorporated in these norm-referenced tests because they may have cultural and gender bias.

## 5. Conclusions

The findings of the present research provide partial support for the a priori hypothesis about the relationships among items of the ball scales. BOT-2 and PERF-FIT ball scales measure a comparable construct of throwing, bouncing, and catching, both including varied complexity of ball items. Both throwing for distance and aiming items seem to belong to separate categories, exhibiting a low relationship with catching items. Throwing for distance requires explosive power and a developed throwing pattern, which can be evaluated by observing the movement pattern. Most ball skill scores differed significantly between 6–7 and 7–8 years, but less between 8–9/10 years; the PERF-FIT discriminated best.

## Figures and Tables

**Figure 1 children-09-00367-f001:**
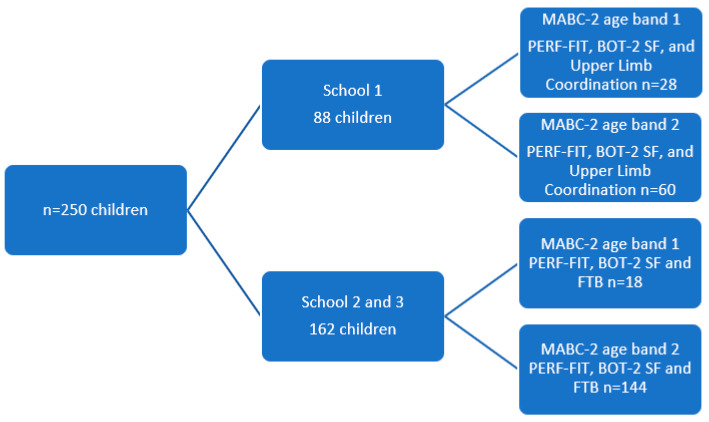
Selection of children and tests administered.

**Figure 2 children-09-00367-f002:**
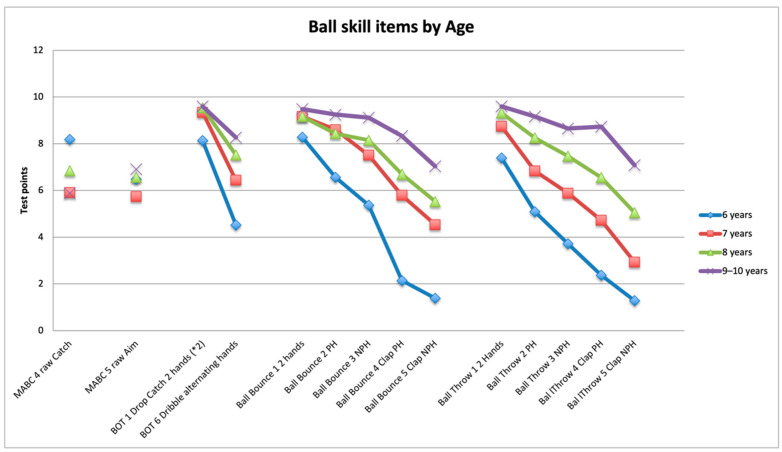
Ball skill items by age. Items of the standardized tests for all 250 children are shown on the *x*-axis, while mean score per age group on those items is depicted on the *y*-axis. The maximum score for all items is 10. PH = preferred hand; NPH = non-preferred hand.

**Figure 3 children-09-00367-f003:**
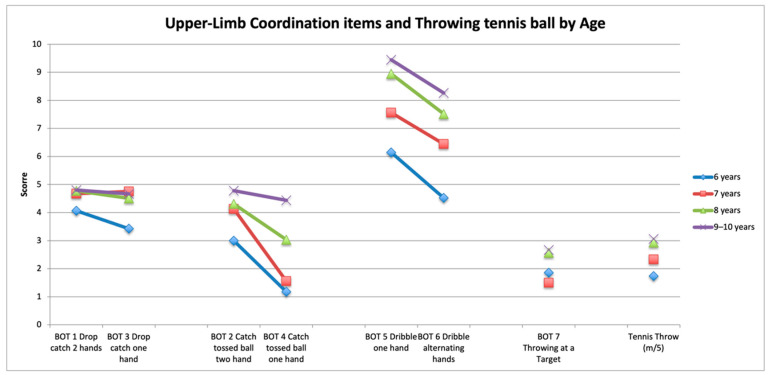
Upper-limb coordination items and throwing tennis ball by age. Items of the standardized tests for the different age groups available for only some of the children (BOT upper-limb coordination subtest *n* = 88 and Fitness Test Battery tennis throw *n* = 162). Tennis distance is presented in total meters divided by 5 to adapt the range.

**Table 1 children-09-00367-t001:** Demographic data of the children included in the study per age band.

		Age Band 1 *n* = 46	Age Band 2 *n* = 204
Boys/Girls	(n/n)	25/21	105/99
Age (years)	Mean (SD)	73.8 (3.3)	100.8 (10.0)
Weight (kg)	Mean (SD)	21.7 (4.5)	28.6 (58.5)
Height (cm)	Mean (SD)	117.3 (6.1)	129.0 (7.5)
BMI (kg/m²)	Mean (SD)	15.6 (2.0)	17.0 (3.5)
MABC-2 (TS)	Mean (SD)	74.0 (13.8)	72.5 (11.8)
≥P25	(%)	69.6	67.6
P16 ≥ x > P5	(%)	17.4	22.5
≤P5	(%)	13.0	9.8

SD = standard deviation; TS = total score of the eight items; P = percentile.

**Table 2 children-09-00367-t002:** Spearman correlation among PERF-FIT, MABC-2, FBT, Fitness Test Battery, and BOT-2.

	PERF-FIT Item 8	PERF-FIT Item 9	PERF-FIT Item 5	BOT-2 UL Item 1	BOT-2 UL Item 6	Overhand Tennis Ball
Spearman’s rho *n* = 250						
PERF-FIT Item 8 Bounce	1	0.76 **	0.46 **	0.44 **	0.62 **	0.44 **
PERF-FIT Item 9 Throw	0.76 **	1	0.51 **	0.37 **	0.61 **	0.47 **
PERF-FIT Item 5 Overhead throw	0.46 **	0.51 **	1	0.18 **	0.32 **	0.50 **
BOT-2 UL Item 1 Drop catch 2 hands	0.44 **	0.37 **	0.18 **	1	0.33 **	0.23 **
BOT-2 UL Item 6 Dribble alternating hands	0.62 **	0.61 **	0.32 **	0.33 **	1	0.48 **
FTB Overhand throw tennis	0.44 **	0.47 **	0.50 **	0.23 **	0.48 **	1
MABC-2 Item 4 raw Catch	0.19 **	0.14 *	−0.03	0.07	0.22 **	0.15
MABC-2 Item 5 raw Aim	0.17 **	0.19 **	0.13 *	0.14 *	0.17 **	0.17 *
MABC-2 Item 4 ISS	0.32 **	0.26 **	0.09	0.12	0.33 **	0.26 **
MABC-2 Item 5 ISS	0.09	0.09	0.04	0.09	0.08	0.07

** Correlation is significant at the 0.01 level (two-tailed); * correlation is significant at the 0.05 level (two-tailed); *n* = 162. PERF-FIT: Performance and Fitness test; BOT-2: Bruininks–Oseretsky Test of Motor Proficiency second edition; UL: upper limb; FBT: Fitness Test Battery; MABC-2: Movement Assessment Battery for Children second edition.

**Table 3 children-09-00367-t003:** Spearman correlation of MABC-2 items, for bands 1 and 2 separately, with PERF-FIT, BOT-2, and FTB.

	PERF-FIT Item 8	PERF-FIT Item 9	PERF-FIT Item 5	BOT-2 ULItem 1	BOT-2 UL Item 6	FTB Overhand Tennis Ball
Spearman’s rho *n* = 46 Age Band 1						
MABC-2 Item 4 raw Catch	0.37 *	0.50 **	0.26	0.24	0.50 **	0.43
MABC-2 Item 5 raw Aim	0.08	0.02	0.23	0.10	0.03	0.14
Spearman’s rho *n* = 204 Age Band 2						
MABC-2 Item 4 raw Catch	0.42 **	0.31 **	0.12	0.17 *	0.36 **	0.25 **
MABC-2 Item 5 raw Aim	0.22 **	0.25 **	0.14 *	0.17 *	0.21 **	0.15

** Correlation is significant at the 0.01 level (two-tailed); * correlation is significant at the 0.05 level (two-tailed). PERF-FIT: Performance and Fitness test; BOT-2: Bruininks–Oseretsky Test of Motor Proficiency second edition; UL: upper limb; FTB: Fitness Test Battery; MABC-2: Movement Assessment Battery for Children second edition.

**Table 4 children-09-00367-t004:** Spearman correlation of BOT upper-limb coordination subtest items with PERF-FIT and MABC-2.

Spearman’s Rho *n* = 88	PERF-FIT Item 8	PERF-FIT Item 9	PERF-FIT Item 5	MABC-2 Item 4 Raw	MABC-2 Item 5 Raw
BOT-2 UL Item 2 Catch tossed ball two hands	0.65 **	0.65 **	0.37 **	0.17	0.21 *
BOT-2 UL Item 3 Drop catch one hand	0.50 **	0.49 **	0.14	0.17	0.15
BOT-2 UL Item 4 Catch tossed ball one hand	0.62 **	0.66 **	0.54 **	0.20	0.17
BOT-2 UL Item 5 Dribble one hand	0.68 **	0.69 **	0.33 **	0.27 **	0.20
BOT-2 UL Item 7 Throwing at a Target	0.37 **	0.39 **	0.19	0.21 *	0.21 *

** Correlation is significant at the 0.01 level (two-tailed); * correlation is significant at the 0.05 level (two-tailed). PERF-FIT: Performance and Fitness test; BOT-2 UL: Bruininks–Oseretsky Test of Motor Proficiency second edition; UL: upper limb; MABC-2: Movement Assessment Battery for Children second edition.

**Table 5 children-09-00367-t005:** Factor analysis for the different ball scales for children 7–10 years of age *(n* = 144). The highest factor loading of an item is shown in bold.

Rotated Component Matrix	Component 1/ Throw and Catch: Ball Handling	Component 2/ Explosive Power	Component 3/Aiming
Explained variance (%)	38.8	13.9	12.7
Eigenvalue	3.10	1.10	1.00
PERF-FIT Item 8 Bounce and Catch	0.81	0.22	0.04
PERF-FIT Item 9 Throw and Catch	0.70	0.33	0.18
PERF-FIT Item 5 Overhead throw	0.01	0.86	0.05
BOT-2 UL Item 1 Drop catch 2 hands	0.68	−0.15	0.01
BOT-2 UL Item 6 Dribble alternating hands	0.79	0.22	0.05
Overhand throw tennis ball	0.29	0.73	−0.01
MABC-2 Item 4 raw Catching	0.53	0.23	−0.44
MABC-2 Item 5 raw Aiming	0.16	0.10	0.90
Extraction method: principal component analysis.			
Rotation method: varimax with Kaiser normalization.			
Age band = 2.00			

Rotation converged in four iterations. PERF-FIT: Performance and Fitness test; BOT-2: Bruininks–Oseretsky Test of Motor Proficiency second edition; UL: upper limb; MABC-2: Movement Assessment Battery for Children second edition.

**Table 6 children-09-00367-t006:** The *p*-values based on Kruskal–Wallis statistics for age group comparison and based on nonparametric post hoc tests to compare differences between two adjacent groups per item.

	Krusal Wallis *p*-Values	Post Hoc Test between 6 and 7 Years *p*-Values	Post Hoc Test between 7 and 8 Years *p*-Values	Post Hoc Test between 8 and 9/10 Years *p*-Values
PERF-FIT Item 8 Bounce	<0.0001	<0.0001	0.110	<0.0001
PERF-FIT Item 9 Throw	<0.0001	0.002	0.001	<0.0001
PERF-FIT Item 5 Overhead throw	<0.0001	<0.0001	0.001	<0.0001
BOT-2 UL Item 1 Drop catch 2 hands	<0.0001	0.011	0.496	0.689
BOT-2 UL Item 6 Dribble alternating hands	<0.0001	0.003	0.034	0.020
MABC-2 Item 4 raw Catch *	0.018		0.006	0.053
MABC-2 Item 5 raw Aim *	0.002		0.007	0.195
Overhand throw tennis	<0.0001	0.052	0.052	0.268

***** Only 7–9 years could be compared. PERF-FIT: Performance and Fitness test; BOT-2: Bruininks–Oseretsky Test of Motor Proficiency second edition; UL: upper limb; MABC-2: Movement Assessment Battery for Children second edition.

## Data Availability

The data presented in this study are available on request from the corresponding author.

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
