# Peer review of "Do We Drop the Ball When We Measure Ball Skills Using Standardized Motor Performance Tests?"

_children, 2022, doi:10.3390/children9030367_

Round 1

Reviewer 1 Report

My recommendations are as follows
Line 27 is not clear this conclusion, I recommend rewriting.
For keywords, I recommend deleting tests and replacing them with other relevant keywords.
Line 33 recommends replacing the word colleagues with collaborators.
Lines 86-87 recommend mentioning the three ball handling models.
I recommend that in section 2.1 the criteria for inclusion and exclusion of the topics included in the research be mentioned.
Table 1: I recommend that the results of MABC-2 (TS) be moved to the results section. Section a2.1 I recommend to mention by age categories and exe demographic data.
Lines 182-183 recommend clarification on the skill level of children.
Lines 266, 270 - I recommend that you assign a relevant title in Figures 2 and 3. Possibly the one in the diagram.
The discussion section assumes that the results of this research are correlated with results from previous studies. In this regard, I recommend extending the discussion section.
Due to age differences and characteristics, this study comes only as a finding, without any intervention. The difference of 4 years between the subjects by comparing the data is not relevant. Some of the subjects had the chance to practice these abilities with the ball during the educational activities and the 6-year-olds did not. Consequently, the premises from which this comparison starts are not relevant.
My recommendation would be to apply a certain exercise program for a certain duration and implicitly to perform an initial and a final test. Then you can compare these skills and make correlations between age categories as evolution.

Author Response

Reviewer 1

We thank the reviewer for the comments.

My recommendations are as follows

  • Comment: Line 27 is not clear this conclusion, I recommend rewriting.

Answer: we have rewritten the conclusion in the abstract. It now reads:

Conclusion: Most ball skill-items in motor tests belong to a comparable construct containing throw, dribble, bounce and catch skills. However, aiming tasks belong to a separate category. Additionally, throwing for distance is a task that requires predominantly explosive power.
2)  Comment: For keywords, I recommend deleting tests and replacing them with other relevant keywords.

Answer: We have added motor performance test to the key words. Naming the tests may be important for meta-analysis searches so we left those in.

  • Comment Line 33: I recommends replacing the word colleagues with collaborators.

Answer: We have changed this as requested.

4) Comment Lines 86-87: recommend mentioning the three ball handling models.

Answer: We have added this to the sentence it now reads: Additionally, there are three different ball-handling patterns to be generated to play the ball, using bare hands, bat or racket.

5) Comment I recommend that in section 2.1 the criteria for inclusion and exclusion of the topics included in the research be mentioned.

The inclusion criteria are mentioned in 2.3 as part of the procedure. “First, we obtained permission from the school district to approach the head teachers. Next, the study purpose, test procedures, benefits and risks were explained to parents. All children between 6 and 10 years of age, in the classes invited for this study, were legible to participate after parents or caretakers signed consent and after the children gave assent. Children whose caregivers answered in the affirmative on any of the questions of the children’s Physical Activity Readiness Questionnaire (PAR-Q) were excluded from the study [29].

6) Comment Table 1: I recommend that the results of MABC-2 (TS) be moved to the results section.

Answer. We agree with the reviewer that MABC-2 data could be described in the results section. However, in this paper the MABC-2 total score is only used to describe the group not as a result for the research questions. Therefore, we opted to leave the descriptive table in the participant section. Additionally, we believe it is easier for the reader have the demographic data next to the flow chart as they are complementary.

7) Comment Section a2.1 I recommend to mention by age categories and exe demographic data.

Answer: We have adapted the table 1 and reported demographics by age categories

8) Comment Lines 182-183 recommend clarification on the skill level of children.

Reply: We have added: Total test time varied between 90-120 minutes depending on the skill level of the children. Children with lower motor proficiency more often needed 2 trials.

9) Comment: Lines 266, 270 - I recommend that you assign a relevant title in Figures 2 and 3. Possibly the one in the diagram.

Answer: We have added the titles to the figures 2 and 3.

10) Comment: The discussion section assumes that the results of this research are correlated with results from previous studies. In this regard, I recommend extending the discussion section.
Answer: We thanks the reviewer for this comment. We discuss the results and relate them to previous studies.

For age, data confirmed what other studies have found [32,33].

For the construct for Ball skills no comparable studies are available.

Based on our findings. we have divided the discussion into 3 paragraphs

  • Constructs measured in the ball scales: here we have no comparable studies, so we give an interpretation of our findings
  • Ball skill competency: here we compare our results to numerous previous studies 34-47
  • Clinical implication, which is mainly an interpretation of our findings with only 1 reference.

11) Comment: Due to age differences and characteristics, this study comes only as a finding, without any intervention.  

Answer: This is a point well taken. This was a correlation study and not an intervention study.

12) Comment: The difference of 4 years between the subjects by comparing the data is not relevant. Some of the subjects had the chance to practice these abilities with the ball during the educational activities and the 6-year-olds did not. Consequently, the premises from which this comparison starts are not relevant.

Comment: With all due respect, we tend to disagree with the reviewer on this point.  If this were true, then no age-related norms could be made in a standardized test for any skill. Given the large sample size, there will be children with more or with less experience in all age groups. Moreover, we did not imply in our text that ball skills are solely a “spontaneous” developing skill. Good ball skills have to be tough/learned. We stated “ball skill performance is known to improve with age and thus with experience, practice, and general development [16,17].”

Moreover, for validity purposes it is important to show that a test is able to discriminate between groups with a known difference. (Called “known group validity”: Cosmin Taxonomy of Measurement Properties).

13) Comment. My recommendation would be to apply a certain exercise program for a certain duration and implicitly to perform an initial and a final test. Then you can compare these skills and make correlations between age categories as evolution.

Answer: We thank de reviewer for this suggestion, and we will definitely keep doing intervention studies because they are dearly needed. However here we have used a cross-sectional design which is standard for a PCA.

Reviewer 2 Report

Thank you for the opportunity to review this manuscript entitled "Do we drop the ball when we measure ball skills using stand- 2
ardized motor performance tests?".

It is a well-prepared manuscript methodologically, but I would like to suggest some questions:

  • Explain if it can influence the results, that more than half of the sample (68.3%) have a P>25.
  • Why have these evaluation batteries been chosen and not others? Justify this.
  • the "n" is always in italics
  • table 6: scores with 0.xxx

Author Response

Reviewer 2

We thank the reviewer for taking the time to review our paper and for the comments (and compliment) provided.

It is a well-prepared manuscript methodologically, but I would like to suggest some questions:

  • Comment: Explain if it can influence the results, that more than half of the sample (68.3%) have a P>25.

Answer: The fact that most children score in the normal range is to be expected in a random group. I would say the fact that 31% of the children were in the “at risk” range is more remarkable. However, for the correlation between outcomes it is good to have some spread in the data (which might be a problem if you select only children below a certain percentile). So we feel including a random sample children over the full range of the motor proficiency spectrum adds to the validity of the findings. We did not select children in anyway on motor performance (as long is parents gave consent and children did not have any underlying condition that might make in unsafe to participate, see questions of the PARQ). We have added this information “Participants in this study were from a low resourced environment and had limited access to organized physical activities and showed a relative high frequency of “at risk motor proficiency” on the MABC-2 classification. Thus, this random cross-sectional sample included children over the full range of the motor proficiency spectrum, which adds to the validity of the findings.”

  • Comment: Why have these evaluation batteries been chosen and not others? Justify this.

Answer: MABC-2 and BOT are the worldwide most frequently used tests. Because these tests do not have norms for African children and because they are too expensive in low resourced areas where we work, the PERF-FIT was developed (we also took away the problem of age bands when developing the test and added power, strength, and agility items).

For the present study, we also wanted to add one more functional throwing item (as opposed to aiming) and found a test that has a standardized description of a throw for distance. Throwing for distance was not included in the PERF-FIT because one of the requirements was that the test could be administered is a small space (We have not added info about the development of the PERF-FIT but have put in a reference). We have added some extra information about the choice of the tests and the PERF-FIT at the reviewer’s request in the introduction and method section. (Highlighted  in yellow)

  • Comment: the "n" is always in italics

Answer: Thank you for pointing this out, it has been adapted throughout the paper

  • Comment table 6: scores with 0.xxx

Answer: This has been replaced by <.0001

Round 2

Reviewer 1 Report

No comments